# Neonatal thyroxine activation modifies epigenetic programming of the liver

Tatiana L. Fonseca 1, Tzintzuni Garcia2, Gustavo W. Fernandes1, T. Murlidharan Nair 3 & Antonio C. Bianco 1✉

The type 2 deiodinase (D2) in the neonatal liver accelerates local thyroid hormone triiodothyronine (T3) production and expression of T3-responsive genes. Here we show that this surge in T3 permanently modifies hepatic gene expression. Liver-specific Dio2 inactivation (Alb-D2KO) transiently increases H3K9me3 levels during post-natal days 1–5 (P1–P5), and results in methylation of 1,508 DNA sites (H-sites) in the adult mouse liver. These sites are associated with 1,551 areas of reduced chromatin accessibility (RCA) within core promoters and 2,426 within intergenic regions, with reduction in the expression of 1,363 genes. There is strong spatial correlation between density of H-sites and RCA sites. Chromosome conformation capture (Hi-C) data reveals a set of 81 repressed genes with a promoter RCA in contact with an intergenic RCA ~300 Kbp apart, within the same topologically associating domain ($\chi^2 = 777$; $p < 0.00001$). These data explain how the systemic hormone T3 acts locally during development to define future expression of hepatic genes.

[1] Section of Adult and Pediatric Endocrinology, Diabetes & Metabolism, University of Chicago, Chicago, IL, USA. [2] Center for Translational Data Science, University of Chicago, Chicago, IL, USA. [3] Department of Biological Sciences and CS/Informatics, Indiana University South Bend, South Bend, IN, USA. ✉email: abianco@deiodinase.org

Thyroid hormone (TH) regulation of gene expression involves binding to nuclear receptors (TR) that are attached or may be directed to TH responsive elements (TREs). This is followed by recruitment of transcriptional coregulators that modify histones and chromatin accessibility to transcriptional enzymes[1]; noncanonical pathways that do not require binding to TREs might also be involved[2]. The genomic actions of TH signal promote localized transition of the chromatin from a tightly folded and less transcriptionally active structure known as heterochromatin to a more loosely folded and active structure known as euchromatin[3–6]. As these processes involve histone modifications, regulation of gene expression by TH is largely fluid, with the rate of transcription of TH-regulated genes fluctuating rapidly according to the levels of the biologically active T3 in the cell nucleus.

In the absence of T3, most unoccupied TRs (uTR) remain attached to TREs, where they form complexes with transcriptional co-repressors to pack the DNA. Examples of uTR-recruited co-repressors include nuclear receptor corepressor 1 (NCoR1) and the silencing mediator of retinoid and thyroid hormone receptors (SMRT; NCoR2), which recruit deacetylases (e.g., HDAC3), methyl transferases (e.g., HMT), and kinases (e.g., protein kinase B [PKB; aka Akt]) to activate facultative heterochromatin formation and inhibit gene transcription[7,8]. Methylation of H3K9, reduction of H3R17 methylation and of phosphorylation/acetylation of H3S10/K14 are all examples of specific histone modifications triggered by uTRs that inhibit transcription[9].

uTRs are known to remain bound to TREs and have repressive effects on gene transcription. Nonetheless, exposure of cells to T3 directs additional TR units to TREs, and shifts TRs association from co-repressors to co-activators; these include steroid receptor coactivator (SRC) family, CBP/p300 or CARM1/SNF5, promoting transcriptional activity. In general, the chromatin modifications induced by T3-TR are the opposite of those caused by uTR, e.g., demethylation of H3K9, methylation of H3R17 and phosphorylation/acetylation of H3S10/K14[9,10]. As a result, the T3-TR not only de-represses but also trans-activates transcription of target genes[1]. T3-TR can also activate gene transcription via long-distance enhancer hyperacetylation, a mechanism that depends on the chromatin context[11].

Less is known about TH regulation of gene expression via DNA methylation, a process that can also result from histone methylation[12,13], and can affect the expression of gene clusters as opposed to a single target gene[14]. In general, the effects of DNA methylation on gene expression last longer when compared to histone modifications, and are involved in developmental transitions such as amphibian metamorphosis and mammalian organogenesis[15], both processes typically sensitive to TH[16,17].

To regulate the timing and intensity of TH signaling on an organ/tissue-specific fashion, developing cells express TH deiodinases, that can both activate (D2) or inactivate (D3) TH. In the embryo, circulating T3 is kept at relatively low levels and tissues predominantly express D3. However, D3 activity diminishes towards birth at the same time that, in some tissues, D2 activity is selectively boosted. A unique blend of D3 and D2 activities seen during this transition independently controls the levels of nuclear T3 in each tissue, hence the timing and intensity of the TH signaling. In the mouse, this process may extend well into the post-natal period given that circulating T3 levels remain low (P1≪P10) through post-natal day 10 (P10)[18,19]. Indeed, peaks of D2-T3 that enhance TH signaling are seen through this period on a tissue-specific basis[3], e.g., embryonic day 17 (E17)-P1 in brown adipose tissue (BAT)[20], or P15 in the cochlea[21].

In the developing liver there is a D2-T3 peak (P1–P2) that briefly doubles T3 content[22] that coincides with the time of

C/EBPa-induced maturation of the bi-potential hepatoblasts into hepatocytes[23]; Dio2 expression in liver is silenced thereafter[22]. This spike in TH signaling is critical for post-natal liver maturation given that liver-specific Dio2 inactivation (Alb-D2KO) was associated with the formation of 1,508 CpG sites of DNA hypermethylation (H-sites) of the adult Alb-D2KO liver genome[22]. As a result, the adult Alb-D2KO liver response to high-fat diet (HFD) is affected[22]; the Alb-D2KO mice exhibit reduced susceptibility to obesity, liver steatosis, hyperlipidemia and alcoholic liver disease[22,24].

Hepatocytes normally undergo post-natal epigenetic reprogramming, including changes in DNA methylation, which are the result of terminal differentiation of hepatocyte precursors[25,26]. For example, between E18.5 until adulthood, ~200,000 CpGs changed DNA methylation by >5%, and ~20,000 CpGs by >30%. These changes in DNA methylation occurred primarily in intergenic enhancer regions and coincided with the terminal differentiation of hepatoblasts into hepatocytes[25,26].

Here, we show that during the last steps of the hepatoblast maturation, a transient surge in D2-mediated T3 signaling interferes with the epigenetic remodeling of hepatocytes, permanently modifying the transcriptome in adult mice. In the absence of this T3 surge, there is a transient (neonatal) increase in H3K9me3 and permanent hypermethylation of discrete DNA areas, which are strongly associated with reduced chromatin accessibility and repression of ~1500 genes in the adult mouse liver. Gene repression involves reduction of chromatin accessibility in distant areas that operate as remote enhancers, located within the same topologically associating domain. These studies explain how D2 locally modifies T3 signaling during development to define future hepatic gene expression.

## Results

Two mechanisms could explain the formation of the 1508 H-sites found in the adult Alb-D2KO liver: (i) defective neonatal DNA demethylation and/or (ii) de novo neonatal DNA methylation, around P1-P5 but before P10, when circulating T3 reached adult levels. To resolve this, we studied the coordinates of the H-sites (Supplementary Table 1) vs. those of thousands of hepatic DNA sites that are normally demethylated during the P1–P21 time frame[25]. Only 25 H-sites were among those that are normally demethylated (Supplementary Fig. 1A), indicating that almost all H-sites in the Alb-D2KO liver resulted from de novo DNA methylation.

**How were the H-sites formed in the Alb-D2KO liver?** Liver-specific Dio2 inactivation reduces T3 content and T3 signaling in the P1-P5 liver[22]. A predominance of unoccupied thyroid hormone receptors (uTR) has been shown to foster a histone environment that has been linked to de novo DNA methylation in a number of developmental settings by recruiting co-repressors that triple-methylate H3K9[9,12,13,16]. Thus, we tested if this could explain the formation of H-sites in the liver of Alb-D2KO by performing a liver ChIP-seq of H3K9me3 in P1, P5, and adult Alb-D2KO mice. Indeed, we found at least 86 H-sites in the Alb-D2KO liver transiently imbedded in H3K9me3-enriched areas that were only present in the Alb-D2KO chromatin (Supplementary Table 1); these sites are hereinafter referred to as HH-sites (Fig. 1a). Sixty-six HH-sites were identified in P1 and 51 in P5; in some cases, H-sites remained surrounded by areas enriched with H3K9me3 throughout the P1 and P5 period. In the adult mice, the differences in H3K9me3 enrichment level between control and Alb-D2KO mostly dissipated, remaining only around 16 of the 86 HH-sites. This dynamic is illustrated in the 20 HH-sites identified on the qA1.2 segment of Alb-D2KO chromosome

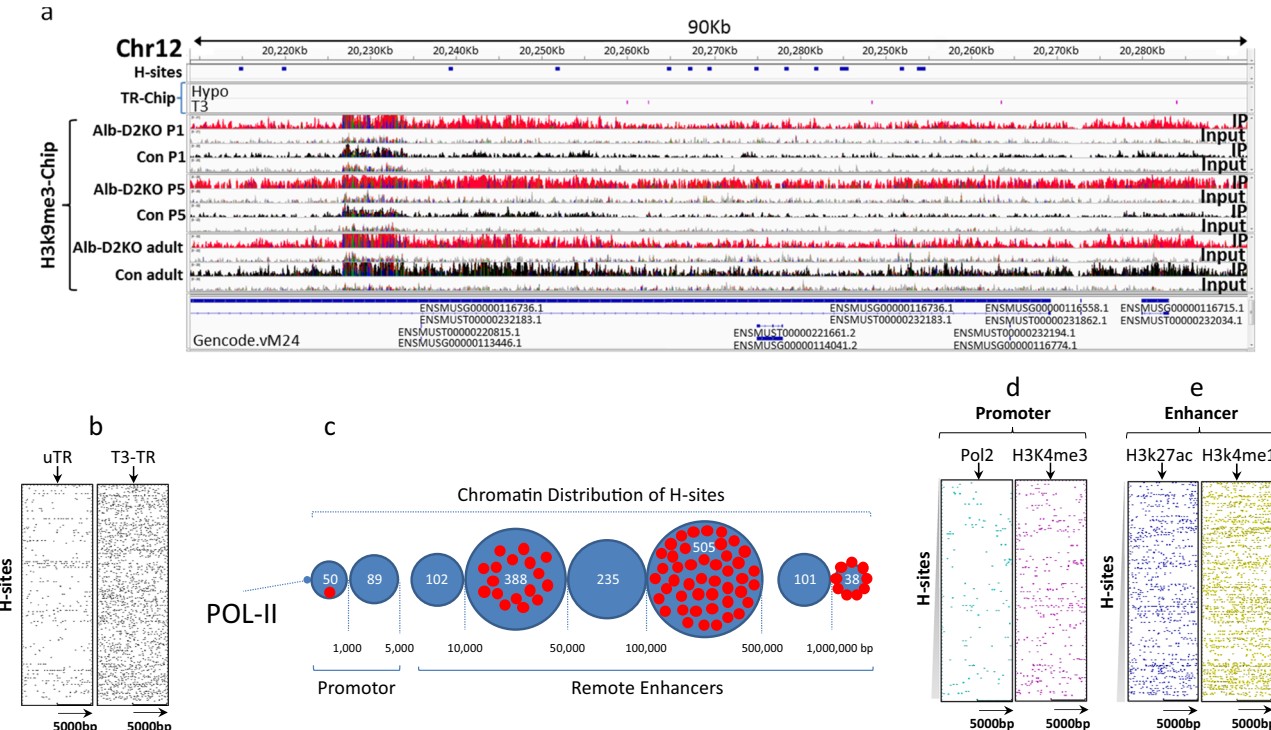

**Fig. 1 DNA hypermethylation sites (H-sites) in the ALB-D2KO mouse liver chromatin. a** browser IGV image of the indicated portion of the mouse chromosome 12 containing 15 H-sites; the following tracks contain ChIP signals for: (top to bottom) H-sites (blue), TR ChIP-seq peaks (hypo and T3-treated - pink), H3K9me3 ChIP-seq peaks in Alb-D2KO (red) and control (black) P1, P5 and adult mice; the input track is shown in gray; a close-up picture of a single H-site is shown in Supplementary Fig. 1B; during optimization of the experimental conditions the pull down was repeated 8 times with different samples; hence replication of experiment (repeating the whole experiment) was not performed as the figure shows data from 12 P1 mice and 6 P5 mice; for adult animals, the figure shows data from 6 mice. **b** heat-maps of H-sites vs. TR ChIP-seq peaks in liver; the x-axis contains the TR peak coordinates at the center +/− 5 Kbp and the y-axis contains the H-site coordinates; the H-site coordinates start at the top and progresses downwards in the following order: Chr 1 through Chr 9, Chr X, Chr 10 through Chr 19; each dot in the heat-map indicates an H-site and its relative distance to a TR peak (center); Hypo indicates TR ChIP peak coordinates obtained from hypothyroid mice, and +T3 indicates TR coordinates obtained from T3-treated mice; **c** distribution of H-sites as a function of polymerase II (pol-II) ChIP-seq peak locations; the numbers at the bottom indicate the distance in base pairs (bp); H-sites are within the *promoter* if up to 1 Kbp; the H-sites are grouped in blue circles according to the distance bracket; the numbers of H-sites in each bracket is indicated inside the blue circles, in white; red dots are H-sites imbedded in areas enriched with H3K9me3 during P1-P5; **d** same as in **b**, except that Pol-II and H3K4me3 (core promoter makers) ChIP-seq peaks were placed at the center of each respective heat-map; **e** same as in **b**, except that H3K27ac and H3K4me1 (enhancer makers) ChIP-seq peaks were placed at the center of each respective heat-map.

12 (Fig. 1a). Within this segment, there was an ~80 Kbp island of intense enrichment of H3k9me3 on P1 and P5; the differences in H3K9me3 dissipated in the adult mice; a smaller 14Kbp island with 5 HH-sites that behave similarly was observed 2 Mbp upstream (Supplementary Table 1).

These results support the idea that during the first few days of life, discrete areas of the Alb-D2KO liver chromatin are enriched with H3K9me3, reflecting an environment that favors the methylation of neighboring DNA sites (H-sites). Here, we captured two snap shots, i.e., P1 and P5, of a process that could last until circulating T3 reached adult levels, i.e. around P10[18,19]. The transient nature of the H3K9me3 pockets and the durability of the H-sites could explain why we only observed their association in 86 cases, given that we only looked at two neonatal time-points.

That liver D2 is only expressed during the first few days of life and the H-sites are only present in the Alb-D2KO livers constitute evidence that formation of H-sites is inherently connected with reduced T3-signaling[22]. Thus, we next wished to define the relationship between H-sites and TRs in the liver. As we were not able to identify high-quality anti-TR antibodies that could be used to successfully prepare a ChIP-seq library in P1 and P5, we reanalyzed two sets of published liver ChIP-seq data obtained with anti-TR antibodies that are no longer available—

one from adult hypothyroid mice, in which uTRs predominate over T3-TRs, and the other from T3-treated adult mice, in which T3-TRs predominate over uTRs,[27] We used these datasets to assess the density of uTRs or T3-TRs in the proximity of H-site coordinates. The resulting heat-maps revealed that while only ~10% of the H-sites were flanked by uTRs in the hypothyroid liver (Fig. 1b), >90% of the H-sites were flanked by T3-TR; T3 has been shown to recruit a larger number of TR units to the TH responsive elements (TRE) (Fig. 1b).

Further assessment of the uTRs and T3-TRs revealed that the sample of 86 HH-sites we identified on P1 and P5 was only flanked by 7 uTRs (within 10 Kbp). In contrast, the same sample of HH-sites was flanked by T3-TRs in 69 occasions (Supplementary Table 1). For example, no uTRs were identified on the qA1.2 segment of Alb-D2KO chromosome 12, or the smaller upstream chromatin area. In contrast, both areas combined presented 9 T3-TRs (Supplementary Table 1). It is thus conceivable that uTRs might not play a decisive role in triggering H-site formation in this setting. Instead, it is more likely that the neonatal D2-mediated enrichment of the liver chromatin with T3-TR sustains recruitment of co-activators and preserves an environment unsuitable for the discrete methylation of H-sites, illustrated by the lower levels of H3K9me3 (Fig. 1a).

### Location of the H-sites in the Alb-D2KO liver chromatin.

To refine the localization of H-sites in liver chromatin, we first annotated the H-sites coordinates and found widespread distribution throughout the genome (Supplementary Data 1). We next reanalyzed published hepatocyte chromatin immunoprecipitation-sequencing (ChIP-seq) data of multiple chromatin markers[28] and contrasted with the H-sites coordinates. Proximity analysis revealed that only few H-sites were located within the core promoter marker Pol-II ChIP peaks; most of the H-sites was found between 5–500Kbp of the Pol-II peaks (Fig. 1c). This distant relationship with core promoters is illustrated in heat-maps in which the coordinates for Pol-II ChIP peaks and an additional core promoter marker H3K4me3 were plotted against the H-sites coordinates (Fig. 1d)[25,29]. The low H-site density in the vicinity of these chromatin markers supports that most are not located in core promoter regions. That most H-sites are likely to be located in remote enhancer regions is suggested by their high density and proximity around the known primed and active enhancer markers, respectively H3K4me1 and H3K27ac (Fig. 1e)[25,29].

### Reduced chromatin accessibility in the adult Alb-D2KO liver chromatin.

Tissue- or cell type-specific DNA hypermethylation is frequently associated with areas of stable heterochromatin, i.e., reduced chromatin accessibility, and has been implicated in the control of gene expression during development[30]. To test if this was case in the Alb-D2KO liver, we used the transposase-accessible chromatin assay followed by high-throughput sequencing (ATAC-seq) of both adult control and Alb-D2KO mouse liver nuclei (Fig. 2a). To generate the ATAC-seq libraries we used high-quality nuclei extracted from frozen adult liver that were quality checked by light microscopy (Fig. 2a). The ATAC-seq libraries yielded the expected fragment length distribution, with the majority of fragments representing areas of inter-nucleosome open chromatin, with progressively fewer large sized fragments representing mono and oligo-nucleosomes (Fig. 2a).

A total of 22,374 areas of open chromatin were identified in the liver of control animals; after gene annotation, almost half of these open areas were placed in promoter regions up to 200 bp of transcription start site (TSS; Fig. 2b). The Alb-D2KO liver exhibited a smaller number of open-chromatin areas, only 14,343, indicating the existence of 8031 regions with reduced chromatin accessibility (RCAs). The reduction of open-chromatin regions detected in the Alb-D2KO included 51% in exons, 53% in introns, and 50% in intergenic areas. There was also a 13% reduction in promoter regions. Of these RCA areas, 1551 were annotated to core promoters and 2,426 to intergenic regions (Fig. 2b). Plotting the RCA coordinates against Pol-II, P300 and H3K4me3 ChIP-seq peaks, revealed high density with variable degrees of superimposition, confirming their proximity to promoter regions (Fig. 2c). At the same time, high density and variable degrees of superimposition features were also observed when RCA areas were plotted against H3k4me1 and H3k27ac ChIP-seq peak coordinates, confirming that they were also present in remote enhancer areas (Fig. 2d).

We next focused on the promoter RCA (p-RCA) areas, and ranked them according to their ATAC-seq signal intensity and signal ratio to two typical chromatin markers H3K4me1 (enhancer) and H3K4me3 (promoter) (Fig. 2e)[29,31]. This approach revealed that 65% of these areas did function as typical

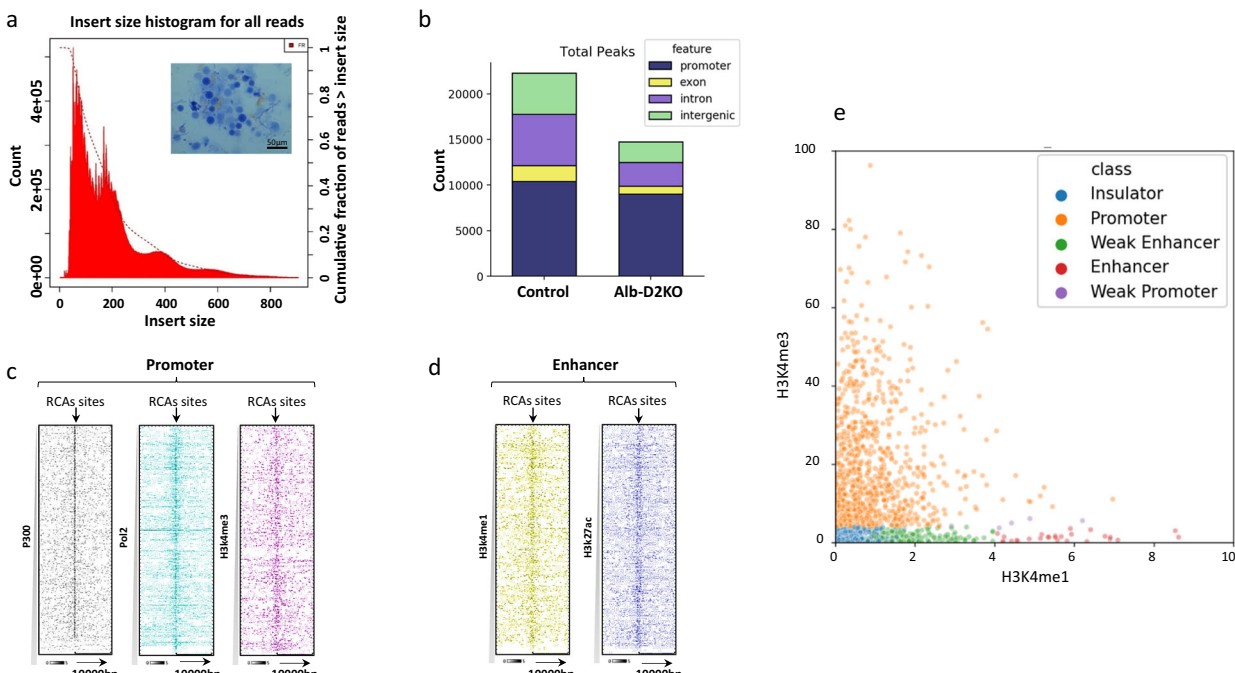

**Fig. 2 Reduced chromatin accessibility (RCA) in the ALB-D2KO mouse liver nuclei. a** the inset contains a representative image of isolated liver nuclei (40x) before adding the transposase enzyme; the graph indicates the lengths of the fragments sequenced within a representative ATAC-seq library; insert size in bp; the first sharp peak (~100 bp) reflects areas of open chromatin, which is followed by 4 peaks of mono (~200 bp) or multinucleosomes of compact chromatin ($n = 4$/group); the picture in the inset is representative of 14 independent nuclei extractions; the graph is representative of two independent library buildouts; **b** genomic distribution of peaks obtained in ATAC-Seq data; eight samples were studied with no further replication; **c** heat-maps prepared as in Fig. 1b, except that the x-axis contains the RCA peak coordinates at the center $+/- 10,000$ bp and the y-axis contains the coordinates for P300, Pol2 and H3k4me3; **d** same as **c**, except that the y-axis contains the coordinates for H3k4me1 and H3k27ac; **e** classification of RCAs according to the relative intensities of H3K4me1 and H3K4me3 ChIP-seq peaks: promoter-RCAs, weak promoter-RCAs, enhancer-RCAs, weak enhancer-RCAs, and insulator-RCAs; each dot represents one RCA area.

core promoters, while 22% were likely to be insulators and 13% local enhancers (Fig. 2e). Of course, these are all functions that can be affected by chromatin accessibility, which is strongly coupled to binding of transcription factors; in most cases binding only occurs when the chromatin is accessible, unpacked[32]. Thus, to identify the transcription factors that were affected by the p-RCA areas in the Alb-D2KO chromatin, we performed de novo motif profiling of these areas using the MEME suite. Motifs belonging to major families of transcription factors were identified, including Sp1–3, E2f3, Klf1, Nrf1, Elk1, and Egr2, many of which are known for playing critical roles in liver development and function (Supplementary Fig. 2)[33–38]. In addition, the second and the third most enriched motifs were for ZNF143 and CCCTC-binding factor (CTCF) (Supplementary Fig. 2). Both are transcription factors that cooperate with the multiprotein complex cohesin to promote chromatin folding, allowing remote enhancers to interfere with transcriptional activity[39].

**Chromatin accessibility and gene expression.** To study gene expression in the Alb-D2KO liver and test whether it was affected by p-RCA areas, we next performed an RNA-seq analysis of adult mouse liver (Supplementary Fig. 1c) and identified 1525 genes differentially expressed in Alb-D2KO mouse (>1.2-fold; $p < 0.05$; Fig. 3a S1D; and Supplementary Data 2). Notably, most of these genes (1363) were downregulated in the Alb-D2KO liver, reflecting the overall reduction in chromatin accessibility. The gene set enrichment analysis (GSEA, Partek Flow) of these differentially expressed genes revealed that the top two gene-sets impoverished in the Alb-D2KO liver were (i) "fatty acid" and (ii) "lipid metabolic process" (ES > 31 and $p < 2.9E-14$) (Supplementary Data 3). In addition, a pathway enrichment analysis (PEA,

Partek Flow) identified metabolic pathways related to drugs and other xenobiotic compounds, as well as five lipid-related pathways among the top 15 pathways impoverished in the Alb-D2KO liver (ES > 5.2 and $p < 0.01$) (Supplementary Table 2).

In order to identify genes in which there was a high likelihood that a p-RCA was affecting its expression directly, we next analyzed the proximity of the 1551 p-RCAs and the 1363 genes with at least one negative RNA-seq site (n-RNA-seq), selecting those that were within 5 Kbp of each other (p-RCA:n-RNA-seq), a configuration that strongly supports a functional relationship between them. This strict criterion led us to 154 p-RCAs (146 genes; Supplementary Data 4), with key roles in liver development and function, including lipid metabolism, mitochondria, redox control, drug metabolism, TGFB-signaling and fibrosis, and cell signaling. Typical examples of these p-RCA:n-RNA-seq areas and the corresponding gene coverages are shown in Fig. 3b.

An analysis of these 154 p-RCA areas revealed that about three-fourths contained putative footprints of transcription factors, including SP1–3 ($n = 63$), E2F3 ($n = 32$), Klf1 ($n = 37$) and NRF1 ($n = 34$) (Supplementary data 4). These footprints were rarely alone; most of the time they were present in groups of 3–4 on the same p-RCA. At the same time, about one-fourth of the p-RCAs was found in insulators. The presence of both cohesin and CTCF (CAC-sites) is typical of insulators that modulate the 3D chromatin structure and gene expression by affecting the connection with remote enhancers. In contrast, insulators that contain cohesin only (CNC-sites) have a local effect, isolating functional promoters and enhancers from spreading neighboring heterochromatin[40]. Using published mouse liver ChIP-seq data[31] for cohesin (Rad21) and CTCF, we looked for overlaps between the 34 RCA/insulator areas and the cohesin/CTCF peaks. There were 30-CAC sites and 4 CNC-sites, highlighting the potential importance of remote regulation

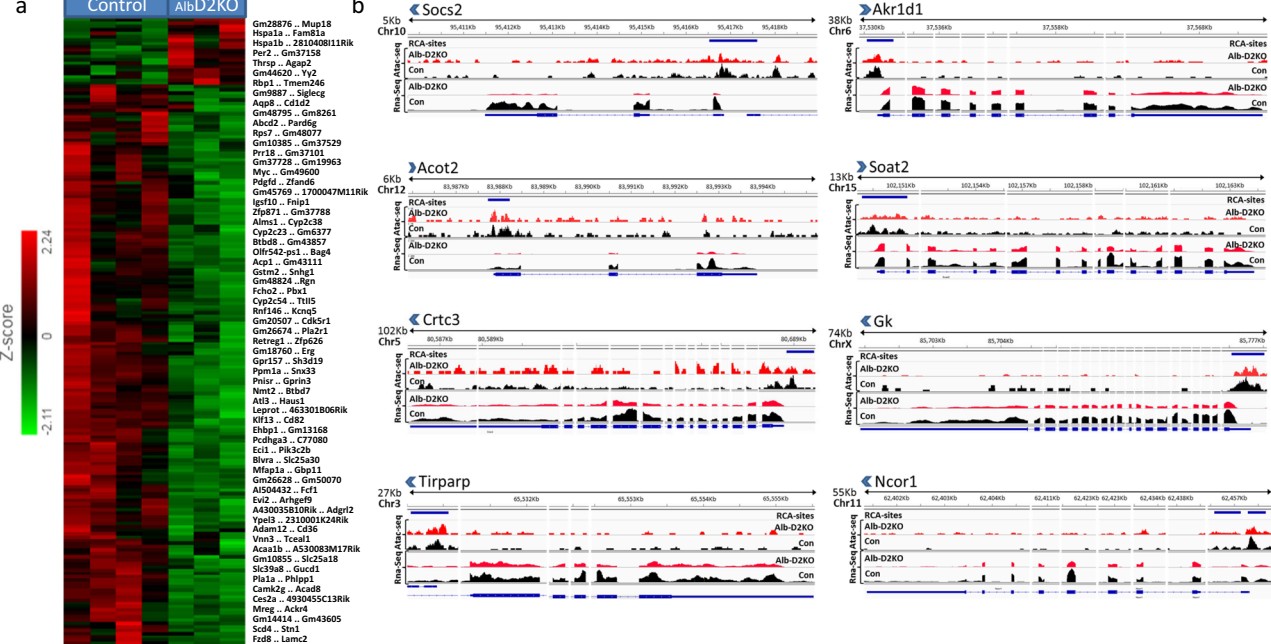

**Fig. 3 Gene expression in the ALB-D2KO mouse liver. a** heat-map of liver RNA-seq data showing clustering of genes affected in the ALB-D2KO mouse; green indicates relative down-regulation; red indicates relative upregulation; fold change is shown on the left; most gene names are indicated on the right (nine samples were studied with no further replication); **b** browser IGV display of transcript areas of selected genes (eight representative genes out of a total of 146 genes) that are downregulated in the Alb-D2KO liver AND exhibit a RCA area within its core promoter region; the distance between the negative RNA-seq peak and the RCA area in all cases is < 5 Kbp; the gene names are indicated on the top of each panel along with the arrow indicating direction of transcription; the red tracks reflects the coverage for the Alb-D2KO Atac-seq and RNA-seq data, and the black tracks reflects the controls; the statistically significant RCA areas are indicated by a blue line.

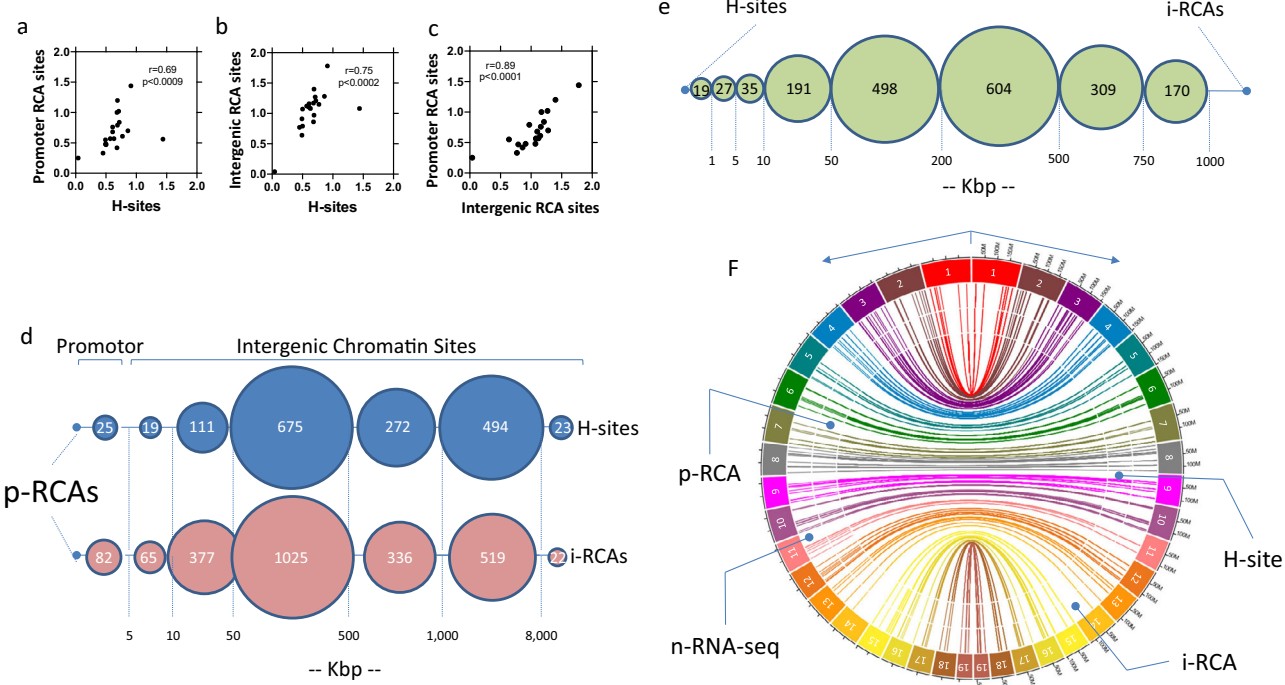

**Fig. 4 Interrelationship between H-sites and RCA areas in the ALB-D2KO mouse liver. a** correlation between the density (number of H-sites/size of the chromosome) of H-sites and promoter RCAs sites in each chromosome; the nonparametric Spearman $r$ and the statistical significance are indicated; no adjustments were made for multiple comparisons [$p < 0.0009$ (two-tailed)]; **b** same as **a**, except that i-RCA sites were used [$p < 0.0002$ (two-tailed)]; **c** same as **a**, except that promoter RCAs and i-RCA sites were used [$p < 0.0001$ (two-tailed)]; **d** distance between H-sites and i-RCA areas vs. p-RCA areas; the number of H-sites and i-RCA areas at each distance bracket is indicated inside the circles; distances are indicated in Kbp; **e** same as **c** except that distances were calculated between intergenic RCA areas (i-RCA) vs. H-sites; there are 572 i-RCA located at a distance >1000 Kbp that are not indicated; **f** Circa representation of the 121 n-RNA-seq areas and p-RCA areas within 1000 bp of *area-1* vs. the closest H-sites and i-RCA to *area-2* areas across the Alb-D2KO liver genome; the outer most ring represents chromosome ideograms; on the right, the size of each chromosome is indicated in Mbp; the inner connecting lines indicate the Hi-C interacting points obtained from ref. [43].

in our model (Supplementary Table 3 and Supplementary Fig. 1e).

**Functional interplay between H-sites, RCA areas and gene expression**. There are multiple pathways through which H-sites could affect gene expression, mainly silencing active promoters or enhancers via creating a p-RCA, or controlling higher-order chromatin structure and the action of long-distance enhancers[30]. Indeed, the analysis of the density of 1508 H-sites and of all 8,052 RCA areas in individual chromosomes revealed a strong positive correlation ($r = 0.85$; $p < 0.0002$). The correlation remained strong when only the densities of H-sites vs. p-RCA areas ($r = 0.69$; $p < 0.0009$; Fig. 4a) or intergenic RCA (i-RCA) areas ($r = 0.75$; $p < 0.0002$; Fig. 4b) (Supplementary Table 4). These findings support our hypothesis of a close, possibly cause-effect relationship, between H-sites and RCA areas.

A more granular analysis of the H-sites coordinates revealed that of all H-sites, only 25 were within 5 Kbp of a p-RCA area (Fig. 4d). Notably, we found that 12 of the 154 *p-RCA:n-RNA-seq* areas did contain at least one H-site within 5 Kbp (10 within 1 Kbp), setting the stage for a possible local negative effect on the promoter of the following genes: 1700047M11Rik, Gstm6, 3110082I17Rik, Mtus1, Dixdc1, Eif5, Dsp, Pde4d, Sco2, Tymp, Odf3b, Cdo1 (Supplementary Data 4). In these locations, the p-RCA areas included mostly core promoters, but also 2 enhancers and 1 insulator (Supplementary data 4).

Most H-sites were much further away from the p-RCA areas. About half of the H-sites was found within 500 Kbp of p-RCAs

and two-thirds up to 1 Mbp away (Fig. 4d). In these cases, any effect on gene expression would need to be remote, probably involving intra- topologically associating domain (TAD) chromatin interaction, which is in keeping with the CAC sites (Supplementary Table 3) and CTCF/ZNF143 footprints detected in p-RCA areas (Supplementary Fig. 2). In the specific case of the remaining 134 *p-RCA:n-RNA-seq* areas, the nearest H-site was found within 50 Kbp of 18 p-RCA sites, between 50 and 500 Kbp of 62 p-RCA sites, and between 500 Kbp and 1 Mbp of 19 p-RCA sites; the remaining 35 p-RCAs were much further away from an H-site (Fig. 4d).

Whereas it is logical to assume that even distant H-sites per se could affect chromatin structure and folding[30], it is also feasible that the disruption of the intergenic enhancers resulted from the combined effects of both H-sites and i-RCAs. Furthermore, distant H-sites could have played a more local role in the formation and maintenance of the i-RCA areas, disrupting proper recruitment of proteins to, and the folding of, chromatin[40]. Indeed, an analysis of the distance between p-RCAs and H-sites or p-RCAs and i-RCAs shows a remarkably similar profile (Fig. 4d), with relatively minor distances between H-sites and i-RCA areas (Fig. 4e), which was not unexpected, considering the dimension of the intergenic environment and the high-order chromatin structure[40]. Nonetheless, it is remarkable that the strongest correlation found was between the density of i-RCA vs. p-RCA areas ($r = 0.89$; $p < 0.0001$; Fig. 4c).

DNA sequences within a TAD are known to physically interact with each other more frequently than with sequences outside the TAD[41]. Thus, we tested whether p-RCA and i-RCA areas

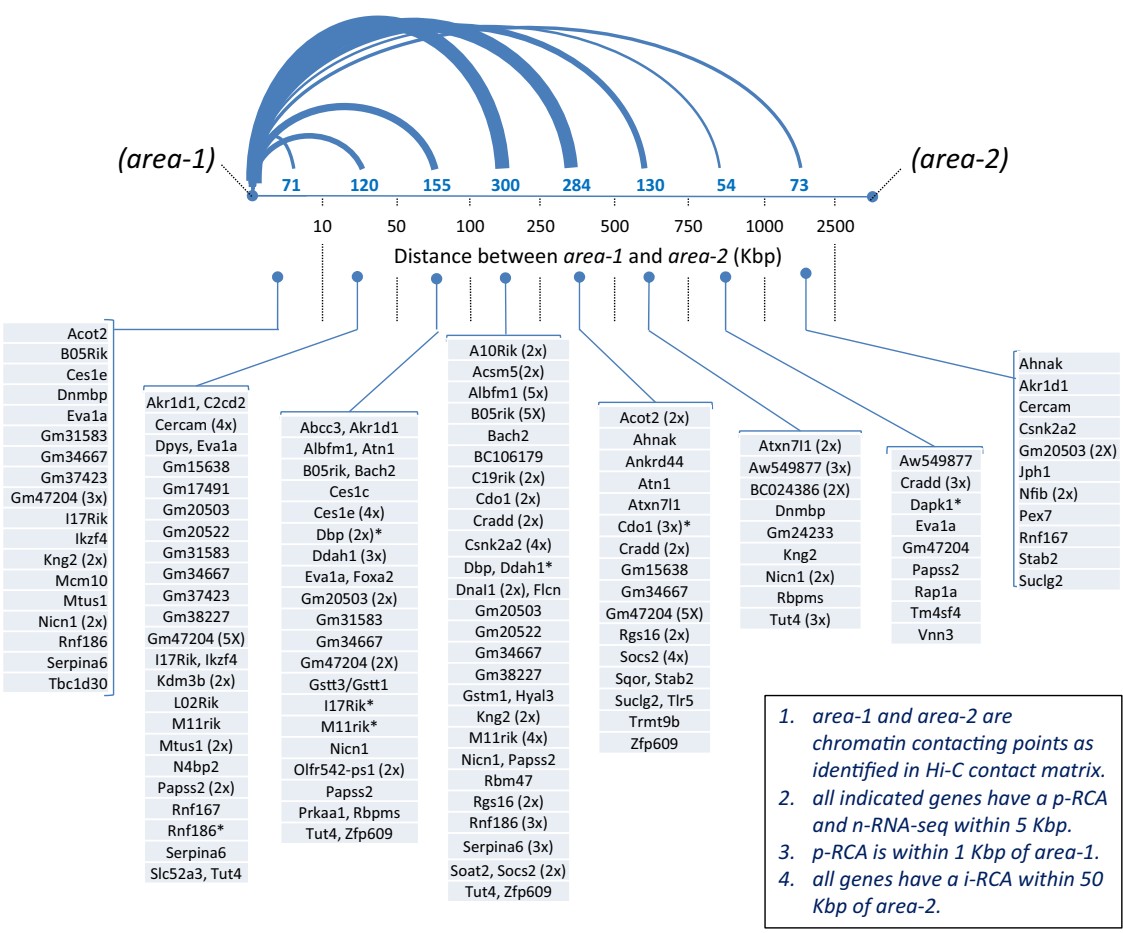

**Fig. 5 Distribution of Hi-C interacting points (*area-1* and *area-2*) located nearby p-RCA areas (*area-1*) and i-RCA areas (*area-2*).** Only *area-1* within 1 Kbp of a p-RCA are shown; the number of *area-2* contacting points at each distance bracket is indicated in green and by the relative thickness of the connecting lines; the 120 gene names shown reflect those genes that satisfy the following criteria: **a** having a n-RNA-seq peak within 5 Kbp of a p-RCA; **b** having a p-RCA area within 1 Kbp of an *area-1*; **c** having an i-RCA area within 50 Kbp of an *area-2*; in parenthesis after each gene name is the number of interacting *areas-2*; Asterisk indicates the presence of a H-site within 50 Kbp of *area 2*. Note that the following genes that were originally included among the 146 are not shown here because they did not meet the strict criterion of having a p-RCA within 1000 bp of a Hi-C *area-1*:Ncor-1, Rmf167, Cdk5R1, Afmid, Sav1, Zbtb1, Ighm, Gm49413, M09Rik, Sco2, Tymp, Odf3b, Dnmdp, Echdc3, Gstm6, Adgrl2, Lurap1l, Gbp11, Smurf1, J07Rik, O19Rik, Pixbc1, Traip, and Acaa1b.

colocalized within the same TADs. We used published TAD coordinates obtained from mouse liver chromatin[31] and used bedops[42] to cross them with the coordinates for all p-RCAs and i-RCAs. The analysis of 3,538 TADs revealed that 472 contained p-RCAs and 677 contained i-RCAs, of which 278 contained p-RCAs and i-RCAs within the same TAD, a highly significant association ($\chi^2 = 557$; $p < 0.00001$). We confirmed this association by manually curating the TAD, p-RCA, and i-RCA data for chromosome #1, with similar results (Supplementary data 5).

We next used chromosome conformation data (Hi-C) obtained from primary liver cell cultures[43] to identify potential chromatin contacting areas that could explain how disruption of a distant enhancer would have affected the 154 *p-RCA:n-RNA-seq* sites and their genes. This was done by first building a contact matrix of all statistically significant interacting pairs separated by a minimum distance of 1 Kbp and a maximum distance of 10 Mbp (*area-1* x *area-2*) for each chromosome. This matrix was analyzed by first mining for coordinates that were within 1 Kbp of the 154 p-RCA sites. We found 988 coordinates and 134 p-RCAs that met this strict criterion; these matrix coordinates were arbitrarily named *areas-1*. The remainder p-RCAs included four genes that contained an H-site near the core promoter, i.e., Sco2, Tymp, Odf3b, Gstm6. Second, we further mined the matrix and verified

that these 988 *areas-1* were in contact with 1060 distant areas, given that some *areas-1* exhibited more than one point of contact; we arbitrarily named these distant points of contact *areas-2*. The average distance between *areas-1* and *areas-2* was ~324 Kbp (1 Kbp–2.4Mbp), well within the constraints of a TAD (Fig. 5). Indeed, for these 146 genes (containing 154-pRCAs), 51 TADs contained only p-RCAs and 131 TADs contained only i-RCAs, whereas 81 TADs contained p-RCAs and i-RCAs within the same TAD, a highly significant association ($\chi^2 = 777$; $p < 0.00001$) (Supplementary Data 6).

To test whether p-RCAs could be directly affected by H-sites via high-order chromatin structure, we circa plotted for each chromosome the 1551 p-RCA vs. the 1508 H-sites, and overlaid the 1060 points of contact provided by the Hi-C analysis (Supplementary Fig. 3a–j). While there was always one or more H-sites nearby an *area-2*, the smallest distance between them averaged ~1 Mbp (Supplementary Data 7). Subsequently, we applied the same rationale for the i-RCAs and verified that they were in general much closer to the 1060 *areas-2*, with the smallest distance between them averaging ~0.53 Mbp (Supplementary Data 7). In addition, in many cases an *area-2* was in the vicinity of more than one i-RCA. For example, an analysis of a sample of all *areas-2* within < 50 Kbp of an i-RCA, identified 273 instances

in which an *area-2* was surrounded by up to 17 i-RCA areas, at an average distance of ~22 Kbp (Supplementary Data 7); as a comparison, there were only 78 H-sites located at < 50 Kbp of an *area-2* (Supplementary Data 7).

This remarkable relative proximity between *areas-1* and *p-RCAs:n-RNA-seq* vs. *areas-2* and i-RCAs & H-sites, as revealed by the Hi-C contact matrix, is illustrated by plotting these elements in a single circa-plot with all chromosomes (Fig. 4f). Of the original 146 genes with a *p-RCA-n-RNA-seq* area, 120 genes were found within 1 Kbp of *area-1*, of which 82 exhibited an i-RCAs within 50 Kbp of 211 *areas-2* (Fig. 5).

## Discussion
The investigation of a dramatic phenotype seen in the Alb-D2KO mouse[22,24] led to the discovery that neonatal inactivation of liver Dio2 reduced local TH-signaling and set in motion a series of events that reduced chromatin accessibility and future expression of key hepatic genes. The present data suggest that a developmental hepatic peak of D2 locally mitigates the typically low postnatal circulating T3 levels of this period of life, which then prevents creation of a nuclear environment—enrichment of discrete chromatin areas with H3K9me3—that could favor the formation of ~1500 sites of de novo DNA hypermethylation (H-sites). These H-sites were mostly distant from core promoters and distributed similarly to the ~1550 p-RCA and ~2400 i-RCA areas; a TAD analysis revealed a highly significant association between these areas within the same TAD. These elements are likely to have disrupted the function of long-distance enhancers, which then failed to interact and activate the core promoter areas, lowering the expression of ~1300 genes involved in different aspects of liver development and function.

The present findings indicate that localized D2-generated T3 plays a role in the terminal maturation of the hepatocytes. This is reminiscent of the role played by D2 in the post-natal development of brown adipocytes and cochlea[20,44]. The timing and intensity of T3 signaling in developing cells is regulated via expression of deiodinases. In the embryo, circulating T3 is kept at relatively low levels and tissues predominantly express the inactivating deiodinase, i.e., D3, limiting exposure to T3. However, D3 activity diminishes towards birth at the same time that, in some tissues, D2 activity is selectively boosted. A unique blend of D3 and D2 activities seen during this transition independently controls the levels of nuclear T3 in each tissue, hence the timing and intensity of the thyroid hormone signaling.

In the case of hepatocytes, the short-lived, localized D2-T3 production, enriches discrete areas of the chromatin with T3-TRs, preventing accumulation of H3K9me3 and possibly the formation of the H-sites. Our H3K9me3 ChIP-seq studies in P1 and P5 Alb-D2KO livers caught the moments during which the chromatin environment could favor formation of H-sites, illustrated by the islands of H3K9me3 surrounding H-site coordinates; different timings could have been involved in the formation of the other H-sites, up until P10, when circulating T3 reached adult levels. Insulin signaling has been shown to play a similar role in the post-natal epigenetic programing of the liver, including differential DNA methylation. For example, changes in the DNA demethylation during the neonatal period were found to be essential for the ligand-activated PPARα-dependent gene regulates the hepatic fatty acid β-oxidation[45]. At the same time, changes in the DNA methylation coincide with the hepatocyte terminal differentiation and occurs after hematopoietic stem cell migration[26].

uTRs have been shown to increase H3K9 methylation through recruitment of SUV39H1[9] and HP1 proteins, a family of heterochromatic adaptor molecules implicated in both gene silencing and supra-nucleosomal chromatin structure[46]. Indeed, the SUV39H1 was also found to cooperate with DNMTs to establish sites of de novo methylation[47], which could explain the formation of the H-sites found in ALB-D2KO liver. Nonetheless, our finding that few uTRs are nearby H-sites suggests that DNA hypermethylation in these areas could be the default hepatocyte differentiation program without the modification normally caused by the D2-mediated surge in T3 and T3-TR. For example, hairless encodes a H3K9 demethylase, and is a highly T3-responsive genes[48].

The reduction in gene expression found in the Alb-D2KO liver is dramatic. There are probably multiple mechanisms that explain such changes, but the ones we found evidence for in the present investigation include (i) disruption of few p-RCA function by local H-sites, (ii) disruption of long-distant chromatin interactions, looping enhancers and promoters, (iii) or a combination of both. The relative proximity and similar distribution of H-sites, i-RCAs and p-RCAs within individual chromosomes is remarkable; it is an indication that indeed most of these elements exist inside the same or neighboring TAD unit, and probably exhibit a functional relationship. These concepts were illustrated in the finding of 81 genes that have a *p-RCA:n-RNA-seq* area along with likely long-distance i-RCA/H-site interacting chromatin areas within the same TAD. It is conceivable that H-sites initiate these processes by creating/maintaining i-RCAs and disrupting the function of the distant enhancers, subsequently depriving core promoters of transcriptional factors that activate gene transcription. Sp1–3, E2F3, Klf1, and Nrf1 were among the potential transcription factors that could be affected in the promoter p-RCAs of the downregulated hepatic genes. It is notable that TR is not one of the transcription factors identified in p-RCAs, confirming that the T3 effects observed on gene transcription are mediated in remote locations.

The analysis of the p-RCAs also revealed negative Nrf1 and Elk1 footprints in the Ncor1 promoter, one of the 146 key downregulated genes with a p-RCA in the adult Alb-D2KO liver. Ncor1 is a TR corepressor[7,8], which in the liver plays an essential role down-regulating T3-signaling. It is fascinating that the liver responds to a reduction in neonatal T3-signalig (due to Dio2 inactivation) and preserves T3-signaling homeostasis by inhibiting the adult expression of a TR corepressor.

In conclusion, the present studies revealed that during the hepatoblast-hepatocyte transition in mice, a short-lived surge in D2 expression and T3-signaling substantially modifies the hepatic transcriptome in adult animals. By acting through intergenic TRs, T3-signaling prevents discrete methylation of specific DNA sites, which would otherwise disrupt the function of areas that operate as remote enhancers. In the presence of the normal neonatal D2-T3 peak, these distant areas interact remotely with dozens of gene promoters, increasing chromatin accessibility and expression of genes involved in multiple hepatic functions. This constitutes a mechanism through which TH regulates gene expression, and explains the critical role played by deiodinases in vertebrate development.

## Methods
**Animals**. Experiments were approved by the University of Chicago Institutional Animal Care and Use Committee - protocol # 72577. C57BL/6J mice with hepatocyte-specific Dio2 inactivation (Alb-D2KO) were obtained by crossing floxed C57BL/6J D2 mice (dio2Flx) with mice expressing Cre-recombinase under the albumin promoter (Cre-ALB) [B6.Cg-Tg(Alb-cre)21Mgn/J; The Jackson Laboratory][22]. The mice were kept at room temperature (22 °C) under a 12-h dark/light cycle, and maintained on a chow diet (3.1 kcal/g; 2918 Teklad Global Protein rodent diet; Harlan).

**Analysis of DNA hypermethylation sites**. Liver hypermethylation sites (H-sites) in the Alb-D2KO mice were obtained from a previous methylome analysis that

used two methods to identify H-sites: (i) Methylation-Dependent ImmunoPrecipitation followed by sequencing (MeDIP-seq) and (ii) Methylation-sensitive Restriction Enzyme digestion followed by sequencing (MRE-seq)[22]. Only the H-sites present in both methods were considered. H-sites were annotated to gencode.vM24. using the Partek Genomic suite software 7.0.

**ChIP-sequencing (seq) and analysis.** Liver ChIP-seq for H3K9me3 was performed in P1, P5 and adult control and Alb-D2KO mice, using the SimpleChIP® Plus Enzymatic Chromatin IP Kit (Magnetic Beads; Cell Signaling). Liver from pups and adult control and Alb-D2KO mice were snap frozen and stored at −80 °C. Four P1 samples ($n = 4$) were studied. Each sample was made up of a pool of 3 livers, totaling 12 mice: two samples were obtained from 6 Cre control animals and 2 samples were obtained from 6 Alb-D2KO animals. Two P5 sample were studied, each made up of a pool of three livers, totaling six mice. One sample was obtained from three Cre control animals and one sample was obtained from three Alb-D2KO animals; three adult livers were processed and sequenced individually. All samples were sequenced twice in different days; replication (repeating the whole experiment) was not performed because we studied 12 P1 mice and 6 P5 mice and results in both groups were similar. So, we considered that the P5 data replicated the P1 data, in what we called neonatal data. Chromatin lysates were prepared, pre-cleared with Protein G magnetic beads, and immunoprecipitated with antibodies against the H3K9me3 (Part Number 13969S- dilution 1:50- Cell Signaling). Beads were extensively washed before reverse crosslinking. Chip-enriched DNA was submitted to the Genome facility at University of Chicago for library preparation and sequence using Illumina NovaSEQ6000. FASTQ files obtained from sequencing were aligned to the Mouse mm10 genome in Partek-flow platform (Partek Inc. v.10.0.21.0509) using the BWA (v.0.7.17). The peaks were identified using MACS2 (v.2.1.1) tool, broad region and $p < 0.05$, compared with respective input samples and annotated using the gencode.vM24. This led us to P1: 280,580, P5: 265,920 and adult: 292,552 peaks in control livers vs. P1: 273,767, P5: 268,693 and adult: 286,124 peaks in Alb-D2KO livers. Areas of H3K9me3 enrichment unique to ALB-D2KO livers were also identified by comparing Alb-D2KO vs. controls samples, which led us to P1: 301,105, P5: 301,098, and adult: 287,032 peaks, which were filtered down to P1: 11,303, P5: 10,005 and adult: 16,730 using −log10($p$-value)>2.

The following published ChIP-seq datasets from adult mouse liver were reanalyzed and also used in the present study: histone markers[28]—GSM722760 (H3K4me1); GSM722761 (H3K4me3); GSM851275 (H3K27ac); GSM722762 (P300); GSM722763 (Pol2); GSM722764 (Input); chromatin-folding markers:[31] Rad21 and CTCF (SRP116021) and TR[27] (SRP055020). The FASTQ files were aligned to transcriptome with BWA (v.0.7.17) using the Partek flow platform (Partek Inc). The peaks were identified using MACS2 (v.2.1.1) tool ($p < 0.05$), compared to respective input samples and annotated using the gencode.vM24. In all instances, pre- and post-alignment QA/QC was done using Partek Flow (Partek Inc).

**ATAC-seq and analysis.** Frozen livers were processed for nuclei isolation using a detergent-free kit (Invent; cat # NI-024). Nuclei integrity was verified under light microscopy (Fig. 2a). Fifty-thousand nuclei were used for the transposase reaction as described[49]. The final libraries were purified using Agencourt AMPure XP beads (Beckman Coulter), and quality checked using a Bioanalyzer High Sensitivity DNA Analysis kit (Agilent); concentration was measured through a qPCR-based method (KAPA Library Quantification Kit for Illumina Sequencing Platforms). Samples were pair-end sequenced with the Illumina HiSeq 4000 platform at the University of Chicago Genomics Facility. The remaining adapter sequences were removed using NGmerge (v.0.3) Sequencing reads were aligned to the Mouse mm10 genome build using BWA (v.0.7.17) and annotated using the gencode.vM24. Sequence quality was assessed using FastQC and tools from the Picard suite including CollectInsertSizeMetrics, which showed an enrichment in short fragments as expected (Fig. 2a). Aligned reads were further filtered to remove chimeric, duplicate, supplementary, and low-quality (MAPQ < 30) alignments. We determined open-chromatin regions (peaks) using Genrich (v.0.6) ($q$-value threshold = 0.05). Run by individual sample, genrich would find only a handful of peaks, which very obviously were not due to ATAC enrichment. These regions were merged across samples into a blacklist of regions and excluded from consideration. Ultimately genrich was run using all samples grouped together while ignoring the blacklisted regions. Differential peaks were identified by overlap subtraction to find open-chromatin regions unique to each condition.

Mouse liver RCAs were classified based on the relative strengths of H3K4me1, H3K4me3, and ATAC signals[31]. Signals were summarized as transcripts per kilobase millions (TPM) values in each RCA region. Those RCA regions with a TPM > 4 for either H3K4me3 or H3K4me1 were segregated into one of three categories, depending on the value of the ratio of H3K4me3 to H3K4me1. High ratios of > 1.5 were classified as a "promoter"; ratios < 0.67 were classified as "enhancer", while intermediate values were classified as "weak promoter." Conversely RCA regions with TPM ≤ 4 in both H3K4me3 and H3K4me1 were segregated into one of two categories depending on the relative values of ATAC-seq to H3K4me1 ratio. If the ATAC-seq signal > H3K4me1 signal, then the RCA was categorized as "insulator". If the ATAC-seq signal < H3K4me1 signal, then the RCA was categorized as "weak insulator".

For the motif analysis, the nucleic acid sequence of promoter ATAC-seq peak regions were extracted and masked using RepeatMasker (v.4.0.7, using pre-defined settings for mouse). Motif analysis was carried out using the meme-chip command from MEME Suite (v.5.0.5) using the MOUSE/HOCOMOCOv10, MOUSE/uniprobe, MOUSE/chen2008, EUKARYOTE/jolma2013, and JASPAR/JASPAR_CORE_2016_vertebrates motif libraries. This multiple analysis tool identified known motifs, de novo motifs, characteristic distances and co-occurrences, and collapsed hits by motif similarity.

**RNA sequencing and analysis.** RNA was isolated from mouse liver using the RNeasy Kit (Qiagen). RNA degradation was monitored using a BioAnalyzer (Agilent). Samples of total RNA with RIN > 7.5 were sent to Genome Technology Access Center at Washington University in St. Louis for library preparation and sequence. Libraries were pair-end sequenced with NovaSeqS4 (Illumina). Base-calls and demultiplexing were performed with Illumina's bcl2fastq software and a custom python demultiplexing program with a maximum of one mismatch in the indexing read. The FASTQ files were aligned to gencode.vM24 transcriptome with STAR (v.2.6.1d) using the Partek flow platform (Partek Inc.). All pre- and post-alignment QA/QC was performed in Partek Flow (Partek Inc.) (Supplementary Fig. 1C). Aligned reads were quantified to annotation model (Partek E/M) and normalized (absolute value). Following the differential analysis (GSA), the biological significance of the changes was interpreted using gene set enrichment analysis (GSEA; Supplementary Data 3) and pathway enrichment analyses (Supplementary Table 2).

**Hi-C data analysis.** Publicly available liver Hi-C datasets (GSE65126) were analyzed using the HICUP[50] pipeline. Briefly, the HICUP (v.0.7.3) pipeline take as input FASTQ data from a Hi-C experiment and produces a filtered set of interaction pairs mapped to the reference genome (mm10). The post-pipeline analysis of the output from HICUP was done using HOMER (v.4.11)[51] to create Hi-C interaction and contact matrices. Matrices were then analyzed for long-range interactions involving p-RCAs, i-RCAs as obtained from the analysis of the ATAC-seq, and H-sites.

**Bioinformatics tools.** R-script using the Bioconductor IRanges package[52] (proximity analysis) was used to obtain the intersection of the H-sites and different chromatin markers on the genomic coordinates for the desired distance. The heatmaps used to analyze proximity between different chromosomes coordinates (Figs. 1b, d, e and 2c, d and Supplementary Fig. 1A), were generated using the interactive platform for analysis Easeq. BED files containing the start/end chromosomal coordinates of each peak were used to generate the images. The circos plots (Fig. 4f and Fig. S2A–J) were created with Circa software (v.1.2.1) tool (http://omgenomics.com/circa). The Integrative Genomics Viewer (IGV) 2.7.2 version was use to visualize and integrated all the genomic data analysis (http://www.broadinstitute.org/igv). The analysis of the topologically associated domains (TADs) and RCA regions (intergenic and promoter) were done using bedops (v.2.4.38)[42]. The results from the bedops were mined for relevant intersections using an R script.

## Data availability

The data that support this study are available from the corresponding author upon reasonable request. The sequencing data generated in the course of this study have been deposited in the GEO under accession number GSE162930. The following publicly available datasets used in the manuscript were downloaded using GEO platform: histone markers—H3K4me1 (GSM722760); H3K4me3 (GSM722761); H3K27ac (GSM851275); P300 (GSM722762); Pol2 (GSM722763); Input (GSM722764); chromatin-folding markers: Rad21 and CTCF (SRP116021) and TR (SRP055020).

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

## Acknowledgements

This work was supported by National Institute of Diabetes and Digestive and Kidney Diseases grants DK58538 and DK65055 to A.B., and NSF #1726218 to T.M.N. We thank Dr. Antonio Lerario for the initial ATAC-seq data analysis, and IUSB for supporting research through faculty research grants.

## Author contributions

T.L.F. conducted all experiments, data analyzes, prepared the manuscript figures and tables; T.G. performed ATAC-seq bioinformatics analysis; T.M.N. performed HiC-seq data analysis and downstream bioinformatics analysis; G.W.F. preparation of heat-maps; A.C.B. planned and directed all studies and manuscript write up.

## Competing interests

A.B. is a consultant for Allergan Inc and Synthonics, Inc.; the other authors have nothing to disclose.
