## [Peer Review File · Nature Communications]

REVIEWER COMMENTS

Reviewer #1 (Remarks to the Author):

Fonseca and colleagues previously reported on a postnatal surge in hepatic Dio2 activity in a mouse model and demonstrated that this T3 signaling was critical for post-natal liver maturation. Liver specific Dio2 inactivation was associated with DNA hypermethylation at multiple sites in the liver genome. In this report they show that the acute increase in T3 signaling permanently modifies hepatic gene expression. Blocking this T3 surge by liver-specific Dio2 knockout results in increased H3K9me3 levels in discrete chromatin areas and hypermethylation of 1,508 DNA sites (H sites) that persisted in the adult liver. These H sites show a strong association with areas of reduced chromatin assembly and reduced gene expression. Chromatin and TH signaling modifies the transcriptome of the adult mouse by preventing DNA methylation at specific sites. This mechanism underlies the observation that T3 acts locally during development to define future chromatin accessibility and expression of specific hepatic genes. These are novel findings of broad interest that demonstrate how short-term thyroid hormone in the neonate results in epigenetic modifications in the adult liver and is consistent with other reported actions of thyroid hormone on chromatin modifications. There are several areas of data clarification that would improve the manuscript, including clarification of cell type identification and how replicates were performed for determination of statistical significance.

1. The tissue studies were performed with whole liver, which contains other cell types in addition to the hepatocytes that were the focus of the study. How were these non-hepatocytes accounted for in the ChIP and RNAseq analysis?
2. The methods for the ChIP sequencing describe 3 livers pooled for P1 and P5. Is the data presented from a single assay? How were replicates of the ChIP assay performed?
3. Figure 1-The text (line 181) states that the ChIP data was obtained with TRapha, but the liver is predominantly TRbeta and the data in reference 27 used the TRbeta antibody. The authors should clarify which data set was used, and if it is data with a TRalpha antibody, should provide analysis with TRbeta.
4. Figure 2B-The overall open chromatin areas were reduced in the liver of Alb-D2KO mice, but was this reduction distributed across all locations or were there any significant differences in the distribution of open sites compared with control?
5. Figure 5A-Would clarify units, shown as "areas" but described in methods as number of RCA or H sites per chromosome.
6. Figure 6-Were there any common characteristics of the gene groups based on the position within the Hi-C interacting points?

Reviewer #2 (Remarks to the Author):

The article from Fonseca et al follows an interesting publication of the same group (Fonseca PNAS 2015) which established that a short pulse of thyroid hormone (TH) occurs in the liver during early mouse post-natal life and has long lasting physiological consequences. The authors already showed that the transient stimulation of immature hepatocytes by TH correlates with a modification of DNA methylation, which is maintained until adult stage.

In the present study, the authors try to understand how the initial effect of TH, mediated by the TR nuclear receptors, eventually lead to a permanent change in the hepatocyte methylation landscape and gene expression pattern. They used again the Alb-D2KO mice in which type deiodinase is selectively eliminated from hepatocytes and in which the early pulse of TH does not take place. They refine the analysis of the liver chromatin of these mice at genome wide scale. More specifically, they perform in both mutant and control mice:

- a time course analysis of an histone tail modification which marks facultative heterochromatin and favor transcriptional repression (H3K9me3 at post-natal day 1, 5 and adult).
- gene expression analysis in adult liver (RNAseq).
- chromatin compaction and DNA accessibility in adult liver (ATAC-seq).

To reach an interpretation, their cross their dataset with available datasets for:

- methylation profiles of mutant and control liver at adult stage, from their previous study (MedipSeq and MRE-seq).
- chromatin occupancy by TR in hypothyroid or hyperthyroid adult liver (ChIPseq)
- Topologically associated domains (TADs) of cultured hepatocytes (HiC).
- chromatin occupancy in adult liver for polymerase 2, Rad21 and CTCF
- several other histone marks (ChIPseq)

The authors then conclude that the long-term consequence of TH stimulation in the neonates liver are the results of at least two mechanisms:

- in Alb-D2KO the continuous presence of unliganded TR on chromatin favors DNA methylation at specific sites. The link remains unclear as there is no spatial correlation.
- the change in DNA methylation alters the later chromatin accessibility to other transcription factors.
- it also changes the long distance interactions between enhancers and promoters in topologically associated chromatin domains.

Overall this is a well conducted study, addressing a very interesting question, which has important physiological consequences. However I am not convinced at this point that the data are fully supporting the authors interpretation, and sometimes I do not understand this interpretation. Overall, there is a lot of novel information. The authors, and the reader, struggle to gain a global view of the situation. More specifically:

- This is an in vivo model, which entails a number of complications:

- 1) In their previous work, the authors indicate that, although most physiological parameters are normal in these mice, they are clearly hyperphagic. This might modify gene expression in the liver, and to introduce some confusion in the data, notably RNAseq data. A possibility would be to equalize feeding, and use qRT-PCR to ascertain that the regimen does not alter the expression of key genes.
- 2) The possibility remains that later episodes of transient Dio2 expression were missed. It would be the best interest of the authors to show that the consequences of a transient increase of TH in liver is not only visible in their transgenic model but also when mice are made transiently hypothyroid at early stages by pharmacological treatment. One expects that the adult liver would be similarly impacted.
- 3) Reciprocally it would be good to restore a normal phenotype in these mutant mice by treating them soon after birth with a single pulse of T3 and show that the adult liver phenotype disappears.
- 4) One should predict that TR mutations have a similar influence on gene expression (gene expression in adult liver of THRA/THRB double KO has been published PMID: 12776178).

- Histone tails methylation.

1) Figure 1A is hardly readable. It would be useful to add a magnified view of a region of interest, as even the existence of peaks is not obvious from the picture. This would help to see if there is close connection between DNA methylation and H3K9me3 modification. Also other areas should be shown as one gets the impression of a uniform increase in H3K9me3 level, while the authors explain that this does occur at very specific locations.

2) Most importantly, it seems that there is already a clear difference at P1 between the level of H3K9me3 of control and mutant liver, occurring before the peak of Dio2 expression. Did the process start earlier?

- Connection between TR and epigenetic modifications.

1) As the authors acknowledge, the simple hypothesis, according to which unliganded TR would recruit DNA methylases at specific genomic locations does not fit with the data. However, Figure 1B is puzzling. As I understand from PMID: 25916672, unliganded TR bind at the same position that liganded TR, however chromatin binding is more stable in presence of hormone, and a larger number of peaks is identified by ChipSeq. Do the authors suggest that the fraction of binding site that are only detected in the presence of T3 in adults are selectively associated to hypermethylated sites? Could it be that at some sites, liganded TR can overcome the negative effect of DNA methylation and bind?

2) The ChipSeq data come from adult liver, whereas the authors are testing a hypothesis of a process, which takes place in neonates. In any case, it seems difficult to make any reasoning without performing a ChipSeq analysis of TR at post-natal day 1.

3) As a local effect of TR is ruled out, the authors make the tempting hypothesis that the early pulse of TH, mediated by TR, has long lasting consequences on chromatin 3D organization. The connection between TR and CTCF, which might modulate insulators function and TADs organization has been suggested before (PMID: 12660164) and the hypothesis has been tested in a neural cell line using ChipSeq data (PMID: 23382204, suppl data). I believe that this could be analyzed more precisely, notably by comparing the results in liver with the results in the cell line.

4) I could not follow the final analysis of the Hi-C data. I understand that site of reduced accessibility to transposase in ATAC-seq (RCA) are often close to hypermethylated sites. If I am correct, the authors suggest that the presence of RCAs within the same TAD of a gene that is down regulated in the mutant mice explains the down-regulation. But I see no statistical evidence for this: are all the genes in the same TAD regulated in the same way? Could the authors used a control set of genes, which are not differentially expressed to test this hypothesis? Of a control HiC dataset coming from another tissue?

5) If the correlation between methylation and RCA is strong (line 296), why are hypermethylated sites colocalize with RCA only seen in 8 out of 146 genes?

6) Motif search (Figure 3) only provides indications, not evidence, for the presence of transcription factors. Here it is likely that a single C-rich motif can be considered either as aSp1, CTCF or Klf1 binding site. I thus believe that this information is of little value, and is in part misleading. I would suggest to place this in supplementary data and to temper the text (line 426 "Sp1-3, E2F3, Klf1, and Nrf1 were among the key transcription factors, which were missing in the promoter..." is a clear over-interpretation of this motif search, as the actual presence of these transcription factors was not addressed).

Other points

Line 181: is alpha a typo?

Line 270: shall we really consider that RNAseq provides "peaks"? is it only down-regulated genes?

Line 433: if NcoR is downregulated, should the liver of mutant mice be hypersensitive to T3? Based on PMID: 32453730 one would predict that specific enhancers would be acetylated, perhaps demethylated, and a set of genes overexpressed.

In conclusion, the authors chose a very ambitious task, which was to elucidate an interesting imprinting process identified in their previous publication. However they did not convince me that they succeeded to at least identify some key steps in a complex series of event. Additional experiments are certainly needed.

REVIEWER COMMENTS

Reviewer #1 (Remarks to the Author):

These are novel findings of broad interest that demonstrate how short-term thyroid hormone in the neonate results in epigenetic modifications in the adult liver and is consistent with other reported actions of thyroid hormone on chromatin modifications.

Thank you.

There are several areas of data clarification that would improve the manuscript, including clarification of cell type identification and how replicates were performed for determination of statistical significance.

1. The tissue studies were performed with whole liver, which contains other cell types in addition to the hepatocytes that were the focus of the study. How were these non-hepatocytes accounted for in the ChIP and RNAseq analysis?

The Cre mouse utilized in these studies provide cell specificity because albumin is only expressed in hepatocytes. So, the inactivation of the Dio2 gene only occurred in hepatocytes. In all experiments, we used Cre littermate mice as controls. Therefore, the interference of other cell types contained in our ChIP-seq and RNA-seq samples was neutralized by the control samples.

2. The methods for the ChIP sequencing describe 3 livers pooled for P1 and P5. Is the data presented from a single assay? How were replicates of the ChIP assay performed?

Four P1 samples were studied. Each sample was made up of a pool of 3 livers. Each sample was sequenced twice in different days. Two samples were obtained from 6 Cre control animals and two samples were obtained from 6 Alb-D2KO animals. Two P5 sample were studied. Each sample was made up of a pool of 3 livers. Each sample was sequenced twice in different days. One sample was obtained from 3 Cre control animals and one sample was obtained from 3 Alb-D2KO animals. This information was clarified under material and methods.

3. Figure 1-The text (line 181) states that the ChIP data was obtained with TRalpha, but the liver is predominantly TRbeta and the data in reference 27 used the TRbeta antibody. The authors should clarify which data set was used, and if it is data with a TRalpha antibody, should provide analysis with TRbeta.

Thank you. We unintentionally caused confusion. α TR, as included in the manuscript, means antibody against TR. We are used with this nomenclature but agree that this is confusing. We now spelled out anti-TR antibodies.

4. Figure 2B-The overall open chromatin areas were reduced in the liver of Alb-D2KO mice, but was this reduction distributed across all locations or were there any significant differences in the distribution of open sites compared with control?

Thank you. We included those comparisons in the manuscript. The reduction of open chromatin regions detected in the Alb-D2KO included 51% in exons, 53% in introns and 50% in intergenic areas. There was also a 13% reduction in promoter regions. This is now included in the results section.

5. Figure 5A-Would clarify units, shown as “areas” but described in methods as number of RCA or H sites per chromosome.

Thank you. We edited the figure and included “sites” instead of “areas”.

6. Figure 6-Were there any common characteristics of the gene groups based on the position within the Hi-C interacting points?

We looked at the 7 groups of genes using the pathway enrichment analysis that is part of Partek Flow but could not find anything unique that would distinguish one group from another.

Reviewer #2 (Remarks to the Author):

The article from Fonseca et al follows an interesting publication of the same group (Fonseca PNAS 2015) which established that a short pulse of thyroid hormone (TH) occurs in the liver during early mouse post-natal life and has long lasting physiological consequences. The authors already showed that the transient stimulation of immature hepatocytes by TH correlates with a modification of DNA methylation, which is maintained until adult stage.

Thank you.

In the present study, the authors try to understand how the initial effect of TH, mediated by the TR nuclear receptors, eventually lead to a permanent change in the hepatocyte methylation landscape and gene expression pattern. They used again the Alb-D2KO mice in which type deiodinase is selectively eliminated from hepatocytes and in which the early pulse of TH does not take place. They refine the analysis of the liver chromatin of these mice at genome wide scale. The authors then conclude that the long-term consequence of TH stimulation in the neonates liver are the results of at least two mechanisms:

- in Alb-D2KO the continuous presence of unliganded TR on chromatin favors DNA methylation at specific sites. The link remains unclear as there is no spatial correlation.

In general, we agree with the idea that there is more to be learned about the mechanistic connection between unliganded TRs and DNA methylation; the reviewer’s concern of “there is no spatial correlation” is puzzling though because one should also take into account that remote unliganded TRs could affect DNA methylation. We could have examined remote chromatin interactions between uTRs and H-sites, but in our view with little added value. Two important facts remain, namely that all changes observed are initiated by a hepatocyte reduction in T3 levels and thus increased uTRs, which leads to formation of H-sites in the hepatocytes.

More could be learned if we performed multiple TR ChIP-seq, at least one each day during P1 and P10. A major problem with this approach is that TR antibodies are not good. We used data from adult mice obtained by Dr. Cheng at the NIH because we were not able to obtain reliable immunoprecipitation with commercially available antibodies, or with antibodies provided by Dr. Shi, also at the NIH. We asked, but Dr. Cheng could not find the antibodies she used to share with us.

Nonetheless, we feel that the trigger for DNA methylation must be a reduction in T3 content in the

hepatocytes caused by Dio2 inactivation. T3 content in the liver drops by 50% (previous PNAS publication). Even though we did not measure, based on what is known about TRs, we hope this reviewer would agree that this drop in T3 content increases the relative presence of unliganded TRs in the cell nucleus. Others have shown in different models that unliganded TRs favor the formation of heterochromatin.

Again, we agree with the reviewer that there is still more to be learned about the molecular details between unliganded TRs and DNA methylation, and we will do this in subsequent studies; the reviewer agrees with us that in the current manuscript “there is a lot of novel information”; we have toned down our conclusions accordingly to satisfy the reviewer.

- the change in DNA methylation alters the later chromatin accessibility to other transcription factors.

- it also changes the long distance interactions between enhancers and promoters in topologically associated chromatin domains.

Overall this is a well conducted study, addressing a very interesting question, which has important physiological consequences.

Thank you.

However I am not convinced at this point that the data are fully supporting the authors interpretation, and sometimes I do not understand this interpretation. Overall, there is a lot of novel information. The authors, and the reader, struggle to gain a global view of the situation. More specifically:

- This is an in vivo model, which entails a number of complications:

1) In their previous work, the authors indicate that, although most physiological parameters are normal in these mice, they are clearly hyperphagic. This might modify gene expression in the liver, and to introduce some confusion in the data, notably RNAseq data. A possibility would be to equalize feeding, and use qRT-PCR to ascertain that the regimen does not alter the expression of key genes.

Thank you. An increased food intake would modify liver gene expression. Typically, this would include an increase in genes involved in anabolic processes, such as PPAR-g, and a decrease in genes involved in catabolic processes. However, as shown in Supplemental Table 4, the GSEA of the RNA-seq indicates the opposite. Several catabolic processes, including fatty acid catabolic process, fatty acid oxidation, lipid oxidation, lipid catabolic process, are enriched in the Alb-D2KO liver. In fact, a reduction in PPAR-g mRNA levels was clearly detected in these animals (Supplemental Table 3) as early as in our PNAS publication.

Table1- Relative Dio2 mRNA levels in P1-P30 mice.

Age	Dio2 expression in liver
P1	1.00±0.79
P3	0.51±0.77
P5	0.36±0.23
P10	0.16±0.019
P17	0.12±0.056
P30	0.00±0.000

D2 mRNA levels relative to 18S mRNA levels and normalized to P1. Values are mean ± SEM of 3 independent samples.

Nonetheless, we went back to our data and identified liver samples obtained from mice (one Cre mouse and one Alb-D2KO mouse) that exhibited similar food intake during a stretch of 4 days (Cre: 3.46±1.49 vs Alb-D2KO 4.15±0.98 g/day) immediately before the animals were killed. These animals also had similar body weight (Cre: 29.7g vs. Alb-D2KO: 31g). As a group the Alb-D2KO mice ate more, but there was

sufficient variation within the group that allowed us to identify these two animals, what precluded us from doing a new time-consuming experiment as suggested by the reviewer. The advantage is that we do have the RNA-seq of both animals, which were re-analyzed separately utilizing the same parameters as the original analysis that included all animals. We found that, despite very similar food intake, 71% of the 146 key genes included in Supplementary Table 6 and Figs. 5f-6, were also identified in this limited analysis.

Thus, we feel it is unlikely that the increased food intake in the Alb-D2KO mice play a substantial role in determining the modification in liver gene expression that we are studying.

2) The possibility remains that later episodes of transient Dio2 expression were missed. It would be the best interest of the authors to show that the consequences of a transient increase of TH in liver is not only visible in their transgenic model but also when mice are made transiently hypothyroid at early stages by pharmacological treatment. One expects that the adult liver would be similarly impacted.

Thank you.

(i) to address the possibility that “later episodes of transient Dio2 expression were missed”, we performed a more detailed screening for Dio2 expression on the liver of WT animals. We measured Dio2 mRNA using RT-qPCR in the liver of P1, P3, P5, P10, P17 and P30 (3 mice for each age). As with the Cre mice, we found the highest levels of Dio2 mRNA on P1, which were followed by a progressive decrease in expression, until not being detectable (table 1).

(ii) we were confused by the second part of your comment: “transient increase of TH in liver is not only visible in their transgenic model but also when mice are made transiently hypothyroid at early stages by pharmacological treatment”. Our understanding is that you meant “transient decrease of TH in liver”.

Considering this to be the case, making P1-P10 mice transiently hypothyroid with pharmacological treatment is not practical, if at all feasible. At this age, the animals drink milk and thus we would have to make the females hypothyroid so that the milk would contain anti-thyroid medication. This would need to start while the females were still pregnant, otherwise how else would we ensure that P1 animals are receiving antithyroid meds. Alternatively, one could inject anti-thyroid meds on P1-P10, but these drugs would require time to act. I can think of other scenarios that could be used, for example, make everybody hypothyroid and treat them with T4 and stop T4 at some point during pregnancy, but none that could be performed without introducing major unintended indirect effects.

Please keep in mind that the Alb-D2KO is not transiently hypothyroid. Their liver is. So, to reproduce this model pharmacologically is just not possible at the moment. By using drugs that inhibit thyroid activity would result in systemic hypothyroidism; not only the liver, but all other organs and tissues would be hypothyroid. This is not our model. Deiodinases act by mediating local control of thyroid hormone action, this is one of the major points of the present investigation.

3) Reciprocally it would be good to restore a normal phenotype in these mutant mice by treating them soon after birth with a single pulse of T3 and show that the adult liver phenotype disappears.

Thank you. The response here is essentially the one given above. A single pulse of T3 will result in systemic thyrotoxicosis. How would we know if the effects observed in the adult mouse were caused by an effect of the pulse of T3 in the liver or somewhere else? Brain, adipose tissue, pancreas, GI tract? By

knocking out Dio2 only in hepatocytes, we create a state of localized hypothyroidism that cannot be easily reconstructed pharmacologically.

4) One should predict that TR mutations have a similar influence on gene expression (gene expression in adult liver of THRA/THRB double KO has been published PMID: 12776178).

Thank you. We respectfully disagree. Because of the way TRs function, mice carrying TR mutations have dramatically different phenotypes when compared to the phenotype developed by TR KO mice. This is because most empty TR molecules remain bound to the DNA and do have an effect on gene expression. In contrast, in the TR KO, there are no TR molecules, thus there is only a minor overall effect on gene expression. In addition, and most importantly, a dominant negative TR mutation for example, will not reproduce our model because in the Alb-D2KO mouse there is an increase in the unliganded TRs during the first days of life that is followed after a few days by normal occupancy of the receptors after P10, when serum T3 levels increase. This does not occur in the knock-in mouse carrying a dominant negative TR allele, such as the PV mouse, for example.

Nonetheless, to satisfy the reviewer, we looked at the PMID: 12776178, and saw a study of T3-responsiveness in wild type and double TR knockout mice. We tried somehow to correlate these data with our data and found that only 1 gene identified by the authors of PMID: 12776178 is present in the list of 146 genes we show in Supplemental Table 6 and Fig. 5f-6. Again, we emphasize that a transient increase in uTRs during P1-P10 (as found in our model) is fundamentally different from the double TR KO model.

- Histone tails methylation.

1) Figure 1A is hardly readable. It would be useful to add a magnified view of a region of interest, as even the existence of peaks is not obvious from the picture. This would help to see if there is close connection between DNA methylation and H3K9me3 modification. Also other areas should be shown as one gets the impression of a uniform increase in H3K9me3 level, while the authors explain that this does occur at very specific locations.

Thank you. A close-up of a single H-site is now provided as Supplemental Figure 1B. When compared to DNA methylation, histone methylation is fluid, not stable over time. That is essentially why we did a ChIP-seq on P1 and P5. Whereas the data was largely similar on both days, there were differences because of the fluidity of the process. In addition, triple methylation of H3K9 is not spatially localized, but spreads across distances of the chromatin in a self-feeding reaction, creating long areas of heterochromatin. The process stops when an insulator is reached, which separates areas of hetero- and euchromatin. This is basically why multiple genes within a single TAD are silenced or activated. Thus, we are puzzled by the reviewer's wish to see "close connection between DNA methylation and H3K9me3 modification". This would require multiple ChIPs with anti-H3K9me3 antibody, on P1-P10, as the maturation of hepatoblasts to hepatocytes is rapidly ongoing at this time. Please look at ref#26, in which the authors identified in the E18-P1 liver about 150 sites of differential methylation; this number increases to about 300 sites between P1-P5, about 750 sites between P5-P10, about 1,200 sites between P10-P15, and about 800 sites between P15-P20. While this is happening, the 1,500 H-sites we identified in the present investigation are being created. Thus, we feel that establishing a spatial connection between DNA methylation and H3K9me3 requires multiple time points with not much added value.

We were fortunate to identify a unique locus on chromosome 12 in which there is a cluster of 15 H-sites, making it easier to establish a connection between density of H3k9me3 and DNA methylation. In

addition, as stated in line 158 of the manuscript, we have identified 86 instances of H-sites imbedded in areas enriched with H3k9me3. If we did more h3k9me3 ChIP-seqs at different times we could have identified more spatial connections between H3k9me3 and H-site, but we feel the point has been made and such an exercise would be beyond the scope of the present investigation.

2) Most importantly, it seems that there is already a clear difference at P1 between the level of H3K9me3 of control and mutant liver, occurring before the peak of Dio2 expression. Did the process start earlier?

Thank you. Most likely as Dio2 inactivation occurs as albumin is expressed (Cre-Alb); this starts at around P15 and increases progressively until the end of hepatocyte maturation.

- Connection between TR and epigenetic modifications.

1) As the authors acknowledge, the simple hypothesis, according to which unliganded TR would recruit DNA methylases at specific genomic locations does not fit with the data.

Thank you. Respectfully, what we propose is that unliganded TRs favors the creation of an environment reach in H3K9me3, which serves as a magnet for DNA methylases. Because of the paucity of unliganded TRs nearby where the H-sites are located, we alternatively proposed that T3-TRs prevent the formation of a pre-programmed environment that is reach in H3K9me3. There are many more T3-TRs nearby H-sites. Perhaps both mechanisms occur at the same time. The issue, as now stated in the manuscript, is that the ChIP for TRs was obtained in adult liver. Thus, without a ChIP for TR done at P1 and/or P5, we will not be able to settle which mechanism is taking place (see discussion above). That is why we say in the manuscript that we cannot exclude that both events are taking place, i.e. (i) uTRs attracting co-repressors leading to DNA methylation vs (ii) T3-TR attracting co-activators preventing DNA methylation. (lines 192 through 201).

However, Figure 1B is puzzling. As I understand from PMID: 25916672, unliganded TR bind at the same position that liganded TR, however chromatin binding is more stable in presence of hormone, and a larger number of peaks is identified by ChipSeq.

Thank you. The publication PMID 25916672 concludes that "...in addition to hormone-independent TR occupancy, ... there is considerable hormone-induced TR recruitment to chromatin..."

Do the authors suggest that the fraction of binding site that are only detected in the presence of T3 in adults are selectively associated to hypermethylated sites? Could it be that at some sites, liganded TR can overcome the negative effect of DNA methylation and bind?

Thank you. No, we do not suggest that. Please see discussion of this issue in lines 192 through 201. The follow-up question is not connected with our work.

2) The ChipSeq data come from adult liver, whereas the authors are testing a hypothesis of a process, which takes place in neonates. In any case, it seems difficult to make any reasoning without performing a ChipSeq analysis of TR at post-natal day 1.

Thank you. We reinforced this important point in the manuscript (line 183), and also discussed this above. However, we think the comparison is informative as it favors a role for T3-TR over uTR in the formation of H-sites in the Alb-D2KO liver.

3) As a local effect of TR is ruled out,

Not clear what the reviewer means by this. The studies are about remote regulation of gene expression by H-sites and reduced chromatin accessibility. We did not study local or remote effects of TR.

the authors make the tempting hypothesis that the early pulse of TH, mediated by TR, has long lasting consequences on chromatin 3D organization. The connection between TR and CTCF, which might modulate insulators function and TADs organization has been suggested before (PMID: 12660164) and the hypothesis has been tested in a neural cell line using ChipSeq data (PMID: 23382204, suppl data). I believe that this could be analyzed more precisely, notably by comparing the results in liver with the results in the cell line.

Thank you. This is an interesting point. Our data indicate that a substantial number of pRCAs is present in insulators. As stated in the manuscript (lines 283-294), there are two main functions for insulators in gene promoters: (i) CAC sites that modulate the 3D chromatin structure and gene expression by affecting the connection with remote enhancers; (ii) CNC sites have a local effect, isolating function promoters and enhancers from spreading neighboring heterochromatin. We used published CTCF and cohesin ChIP-seq data to distinguish between the two types of insulators. We are not proposing that TRs or T3 signaling affect CTCF or TAD organization. We did not look at these processes. In addition, I am afraid that using C17.2 neural cell line stably expressing TRs might not provide a useful comparison from our neonatal mouse liver model.

4) I could not follow the final analysis of the Hi-C data. I understand that site of reduced accessibility to transposase in ATAC-seq (RCA) are often close to hypermethylated sites. If I am correct, the authors suggest that the presence of RCAs within the same TAD of a gene that is down regulated in the mutant mice explains the down-regulation. But I see no statistical evidence for this: are all the genes in the same TAD regulated in the same way? Could the authors used a control set of genes, which are not differentially expressed to test this hypothesis? Of a control HiC dataset coming from another tissue?

Thank you. We did follow the reviewer's suggestion and tested the hypothesis that the coexistence of p-RCA and i-RCA in the same TAD is not random. We first obtained published data containing the coordinates for all TADs in the mouse liver chromatin. Next, we manually studied the coordinates for all p-RCAs and i-RCAs corresponding to the 146 genes studied in Supplemental Table 6 and Fig. 4F-5, and used Chi-square to assess their association. Indeed, the association between i-RCAs and p-RCAs in the same TADs was highly significant (Table 2). Next, we repeated this procedure for chromosome #1, with similar results (Table 3). Lastly, we wrote a program in R and applied the same concept for the whole genome, and found similar results (Table 4).

Table 2- Chi-Square calculation from the TADs that contains P-RCA and I-RCA at 146 genes that contains p-RCA:n-RNA-seq.

	I-RCA (+)	I-RCA (-)	Marginal Row Total
P-RCA (+)	81 (7.63) [706.08]	51 (124.37) [43.29]	132
P-RCA (-)	131 (204.37) [26.34]	3407 (3333.63) [1.62]	3538
Marginal Column Totals	212	3458	3670 (Grand Total)

The chi-square statistic is 777.3221. The p-value is < 0.00001. Significant at p < .05. The chi-square statistic with Yates correction is 766.7643. The p-value is < 0.00001. Significant at p < .05.

Table 3- Chi-Square calculation from the TADs that contains P-RCA and I-RCA at chromosome 1.

	I-RCA (+) in Chr1	I-RCA (-) in Chr1	Marginal Row Total
--	-------------------	-------------------	--------------------

P-RCA (+) in Chr1	26 (12.47) [14.67]	19 (32.53) [5.63]	45
P-RCA (-) in Chr1	48 (61.53) [2.97]	174 (160.47) [1.14]	222
Marginal Column Totals	74	193	267 (Grand Total)
The chi-square statistic is 24.4147. The p-value is < 0.00001. Significant at p < .05. The chi-square statistic with Yates correction is 22.6434. The p-value is < 0.00001. Significant at p < .05.			

Table 4- Chi-Square calculation from the TADs that contains P-RCA and I-RCA in the whole genome.

	I-RCA (+)	I-RCA (-)	Marginal Row Total
P-RCA (+)	278 (90.32) [390.01]	194 (381.68) [92.29]	472
P-RCA (-)	399 (586.68) [60.04]	2667 (2479) [14.21]	3066
Marginal Column Totals	677	2861	3538 (Grand Total)
The chi-square statistic is 556.544. The p-value is < 0.00001. Significant at p < .05. The chi-square statistic with Yates correction is 553.5826. The p-value is < 0.00001. Significant at p < .05.			

5) If the correlation between methylation and RCA is strong (line 296), why are hypermethylated sites colocalize with RCA only seen in 8 out of 146 genes?

Thank you for catching this. This was confusing and the numbers were incorrect. We edited the sentence to clarify and updated the numbers. The correlation between H-sites and i-RCA is strong (Fig. 4B and 4D) but when we restrict the analysis to only those H-sites and i-RCA within 50Kbp of each other, the correlation is weak. This indicates that chromatin folding is likely to be involved.

6) Motif search (Figure 3) only provides indications, not evidence, for the presence of transcription factors. Here it is likely that a single C-rich motif can be considered either as aSp1, CTCF or Klf1 binding site. I thus believe that this information is of little value, and is in part misleading. I would suggest to place this in supplementary data and to temper the text (line 426 “Sp1-3, E2F3, Klf1, and Nrf1 were among the key transcription factors, which were missing in the promoter...” is a clear over-interpretation of this motif search, as the actual presence of these transcription factors was not addressed).

Thank you. Done as suggested.

Other points

Line 181: is alpha a typo?

Yes. Corrected. Thank you.

Line 270: shall we really consider that RNAseq provides “peaks”? is it only down-regulated genes?

Thank you. We replaced “peaks” for sites.

Line 433: if NcoR is downregulated, should the liver of mutant mice be hypersensitive to T3? Based on PMID: 32453730 one would predict that specific enhancers would be acetylated, perhaps demethylated, and a set of genes overexpressed.

Thank you. Based on the microarray studies (PNAS publication) and the current RNA-seq studies, we did not identify changes in the expression of T3-responsive genes in the adult Alb-D2KO liver. T3-responsive genes in the liver are usually regulated by multiple factors; in addition, there are other elements involved in TR function that also affect signaling through this pathway. Future experiments could be done to clarify this interesting point.

In conclusion, the authors chose a very ambitious task, which was to elucidate an interesting imprinting process identified in their previous publication. However they did not convince me that they succeeded to at least identify some key steps in a complex series of event. Additional experiments are certainly needed.

It is puzzling the reviewer feels this way given that the reviewer indicates that “overall this is a well conducted study, addressing a very interesting question, which has important physiological consequences.” and that “overall there is a lot of novel information”. This is a rapidly expanding field and we cannot possibly answer all questions in one study. As the reviewer suggests, we will continue with experiments in this area but they cannot possibly be included in one manuscript.

REVIEWERS' COMMENTS

Reviewer #1 (Remarks to the Author):

The authors have responded to the reviewers comments, appropriately modified the manuscript and provided additional data. These are novel findings that demonstrate how short-term thyroid hormone in the neonate results in epigenetic modifications in the adult liver and is consistent with other actions of thyroid hormone on chromatin modifications.

Reviewer #2 (Remarks to the Author):

I appreciate that some of the affirmations of the manuscript were tempered in the revised version. I still believe that treating Alb-D2KO mice with T3, to restore the post-natal pulse, would be informative and I see a contradiction in the authors position. In their rebuttal the authors write: " A single pulse of T3 will result in systemic thyrotoxicosis. How would we know if the effects observed in the adult mouse were caused by an effect of the pulse of T3 in the liver or somewhere else?" while the abstract says "T3 acts locally during development to define future expression of hepatic genes". So the reasoning remains "cell autonomous". Even if complications are always possible, a post-natal pulse of T3 should revert at least in part the effect of the AlbD2KO. It should notably impact the TADs organization.

I hope that the authors will consider this experiment in the future, to complete a study which contains already very interesting novel information.

REVIEWERS' COMMENTS

Thank you very much to both reviewers. Your comments during the first round of reviews were very helpful and improved the manuscript.

Reviewer #1 (Remarks to the Author):

The authors have responded to the reviewers comments, appropriately modified the manuscript and provided additional data. These are novel findings that demonstrate how short-term thyroid hormone in the neonate results in epigenetic modifications in the adult liver and is consistent with other actions of thyroid hormone on chromatin modifications.

Thank you so much. We are very excited with the data.

Reviewer #2 (Remarks to the Author):

I appreciate that some of the affirmations of the manuscript were tempered in the revised version.

Thank you very much.

I still believe that treating Alb-D2KO mice with T3, to restore the post-natal pulse, would be informative and I see a contradiction in the authors position. In their rebuttal the authors write: " A single pulse of T3 will result in systemic thyrotoxicosis. How would we know if the effects observed in the adult mouse were caused by an effect of the pulse of T3 in the liver or somewhere else?" while the abstract says "T3 acts locally during development to define future expression of hepatic genes". So the reasoning remains "cell autonomous".

Thank you. We apologize for not being clear. The data clearly indicates that the effects of D2-T3 are isolated and localized in the hepatocyte. But please note that while the D2-T3 is acting in the hepatocytes the serum T3 levels remain stable. Thus, the D2-T3 in the hepatocytes is not acting systemically. By giving an injection of T3, as proposed by the reviewer, we will increase the T3 levels in the circulation and so all tissues will be exposed to the increased T3 signaling. Thus, whatever we find in the liver, it will be the combined result of T3 acting in the hepatocyte plus T3 acting in the whole body. If we do this and find that we can revert the epigenetic changes observed in the Alb-D2KO mouse, I guess this will be the expected result. However, if we cannot revert these changes it could be because of indirect interference of other tissues, and not because our hypothesis is flawed.

Even if complications are always possible, a post-natal pulse of T3 should revert at least in part the effect of the AlbD2KO. It should notably impact the TADs organization. I hope that the authors will consider this experiment in the future, to complete a study which contains already very interesting novel information.

Thank you. Yes, we will do the experiment hoping that we will rescue the Alb-D2KO phenotype.